# Effects of an Asthma Education Camp Program on Quality of Life and Asthma Control among Thai Children with Asthma: A Quasi-Experimental Study

**DOI:** 10.3390/healthcare10081561

**Published:** 2022-08-18

**Authors:** Sirasuda Sommanus, Raweerat Sitcharungsi, Saranath Lawpoolsri

**Affiliations:** 1Department of Pediatric, Taksin Hospital, Bangkok 10600, Thailand; 2Department of Tropical Pediatrics, Faculty of Tropical Medicine, Mahidol University, Bangkok 10400, Thailand; 3Praram 9 Hospital, Bangkok 10310, Thailand; 4Department of Tropical Hygiene, Faculty of Tropical Medicine, Mahidol University, Bangkok 10400, Thailand

**Keywords:** asthma education camp, pediatric asthma, asthma-related quality of life, asthma control, knowledge, attitudes and practice survey

## Abstract

Caregiver knowledge and management ability can improve asthma control and quality of life (QoL) among children with asthma. A quasi-experimental study was proposed to assess the effect of a 1 day educational camp program on the QoL of children with asthma and on their caregivers’ asthma knowledge and management. Children with asthma and their caregivers were invited to attend a camp. The Pediatric Asthma Quality of Life Questionnaire (PAQLQ), Childhood Asthma Control Test score, and forced expiratory volume in 1 s were assessed in children at the first, 3 month, 6 month, and 1 year visits. The caregiver’s knowledge, attitudes, and practice (KAP) survey was assessed at each visit. A total of 212 patients were enrolled (mean age: 8.56 ± 1.63 years) but only 72 patients attended the camp. There was no significant difference in baseline characteristics, asthma severity, or asthma risk factors between camp attendees and non-attendees. The KAP of caregivers who attended the camp was significantly higher than non-attendees at the 3 month and 6 month visits (16.86 ± 2.3 vs. 15.95 ± 2.78 (*p* = 0.009); 17.25 ± 2.22 vs. 16.7 ± 2.68 (*p* = 0.04)). QoL did not significantly differ between patient attendees vs. non-attendees. PAQLQ mean score correlated with asthma control, indicating that patients with well-controlled asthma had better QoL than those with unstable asthma control (*p* < 0.001). An asthma education camp can help increase self-management knowledge, even though its effect may be short-term. Integrating asthma education into routine care could enhance asthma management in children.

## 1. Introduction

Asthma is a common and persistent airway inflammation that burdens patients, their families, and healthcare systems. Asthma is a global public health issue that affects 1% to 18% of the population of various countries [1]. According to the World Health Organization, more than 339 million individuals had asthma in 2016 [2]. The Global Asthma Network reported in 2018 that approximately 8–10% of Thai children have asthma, with the prevalence of severe wheeze at 2% [3]. Asthma in children can lead to hospitalizations and school absenteeism, in addition to a negative effect on patients’ self-esteem and emotional wellbeing. The effect of asthma on patients’ physical activity and emotional function can be determined by measuring the asthma-related quality of life (QoL) [4]. Individuals with well-controlled asthma have greater asthma-related QoL. Comorbid conditions (allergic rhinitis and obesity), environmental tobacco smoke, and poor medication adherence are associated with uncontrolled asthma [5,6].

Asthma-related QoL assessment has several benefits, including facilitation of treatment monitoring, measurement of medical status, better understanding of patients’ feelings, and improvement of communication between physicians and patients [7]. The Pediatric Asthma Quality of Life Questionnaire (PAQLQ) was developed and validated by Juniper et al., and it has been translated, validated, and used to assess QoL in children with asthma in several countries [8]. The Thai version of the PAQLQ, which was tested for reliability and validated by Poachanukoon et al., was used to assess QoL in children with asthma in Thailand [9].

Previous studies have demonstrated that educational programs for asthmatic patients and their caregivers can change behavior, enhance knowledge, and improve asthma management skills [10,11]. The educational programs generally provide asthma-related knowledge such as pathology of asthma, medication use, and how to prevent an asthma attack [12]. The good attitude toward asthma management can increase confidence, drug adherence, and a positive relationship with physicians; this could have an impact on the self-management practice to prevent an asthma attack [13,14]. These effects can improve pulmonary function, reduce school absenteeism, reduce days of restricted activity, and reduce the number of emergency room visits. However, little is known about the long-term effects of asthma education camps on asthma knowledge and QoL in settings where caregivers have low socioeconomic status and low education. The aim of this study was to evaluate the long-term effect (12 months) of a 1 day educational camp for children with asthma and their caregivers on patient QoL and caregiver asthma knowledge and management.

## 2. Materials and Methods

### 2.1. Study Design and Population

A quasi-experimental study design was used to compare the QoL of asthmatic children and caregiver asthma knowledge, attitudes, and management in individuals who attended and those who did not attend a 1 day educational camp. The study was conducted from March 2018 to March 2019 at a pediatric allergy unit of Taksin Hospital in Bangkok, Thailand. Children aged 7–14 years who had recently been diagnosed with asthma were enrolled. Asthma diagnosis was based on clinical symptoms and the Global Initiative for Asthma (GINA) guideline. Patients with underlying illnesses such as other chronic lung diseases, central nervous system diseases, cardiovascular diseases, and other chronic illnesses were excluded from the study. Patients and their caregivers received an invitation to attend an asthma education camp. Attendance was voluntary and did not affect routine treatment. At 1–4 weeks after the first enrollment, a 1 day asthma education program was held. Participants who attended the asthma education camp were classed as the intervention group. Patients and their caregivers received education about asthma awareness and management skills at the camp. The education included theoretical and practical aspects of asthma knowledge. Practical teaching tools included group discussion, practice with an inhaler device, and games. Asthma patients who declined to attend the camp served as the control group. Caregivers and children provided their informed consent for study participation.

### 2.2. Data Collection

Patients’ demographic and clinical characteristics were recorded. At baseline, all participating children received a forced expiratory volume in 1 s (FEV1) test, asthma severity assessment using the GINA guideline, and QoL assessment using the PAQLQ and childhood asthma control test (C-ACT). Their caregivers were asked to complete a knowledge, attitudes, and practice (KAP) questionnaire at the baseline visit. The 1 day educational camp was conducted within 1–4 weeks following the baseline visit. Then the intervention group joined the camp to acquire skills in asthma knowledge and management. PAQLQ, C-ACT scores, FEV1, and caregiver KAP scores were reassessed for all participants at the 3 month, 6 month, and 12 month visits. The Thai version of the PAQLQ is a 23-item self-report questionnaire that assesses the functional issues that affect asthmatic children [9]. It comprises items in three domains: symptoms, activity limitations, and emotional function. Responses are on a seven-point scale ranging from 1 (not at all) to 7 (always). The 25-item KAP is a self-administered asthma knowledge questionnaire for parents and/or caregivers of asthmatic children [15]. The version used in this study was adapted from questionnaires used in previous studies [16,17]. The KAP items are divided into four sections: disease information, triggers, treatment, and asthma exacerbation. The Thai version of the C-ACT was used to assess childhood asthma control. This seven-item questionnaire comprises two parts. The first four questions are answered by children, and the last three questions are answered by their caregivers. The C-ACT assesses daytime and nighttime asthma symptoms, use of relief medication, and limitation of daily activities in the previous 4 weeks. C-ACT scores > 23 indicate controlled asthma, and scores <18 indicate uncontrolled asthma [18]. Spirometry was performed according to American Thoracic Society standards [19].

During the 1 year study period, all patients received care from only one pediatric physician. Management of asthma followed routine clinical practice. Asthma severity was assessed according to the GINA guideline. Appropriate medication was given to reduce asthma symptoms and minimize the risk of asthma-related exacerbation. In each patient’s follow-up appointment, step therapy was applied according to the GINA guideline.

### 2.3. Statistical Analysis

Data on demographics and clinical characteristics of enrolled patients were described and compared between asthma patients who attended and those who did not attend the asthma education camp, using Student’s *t*-test or the chi-square test as appropriate. The mean differences in PAQLQ score, FEV1 value, C-ACT score, and caregiver KAP score were calculated and compared between the two groups at each visit. Generalized estimating equations were used to adjust for repeated measures at baseline, 3 months, 6 months, and 12 months. The *p*-values for the overall change in PAQLQ score, FEV1, C-ACT score, and caregiver KAP score were also calculated. In addition, caregiver KAP score was separately compared according to disease information, triggers, treatment, and asthma exacerbation. A *p*-value < 0.05 was considered statistically significant. Analysis was performed using SAS software version 9.4 (copyright© 2021, SAS Institute Inc., Cary, NC, USA).

## 3. Results

### 3.1. Participant Characteristics

A total of 212 children with asthma were enrolled; the mean age was 8.56 ± 1.63 years. Of these, 131 patients (61.8%) were male. The mean age at asthma diagnosis was 5.97 ± 2.26 years, and that at asthma onset was 4.35 ± 2.51 years. Most children had moderate persistent asthma (112; 52.8%), followed by mild persistent asthma (68; 32.1%) and severe persistent asthma (32; 15.1%). The most common comorbidities were allergic rhinitis (46.7%), snoring (24.1%), and atopic dermatitis (20.8%). Potential risk factors for atopy were aeroallergen sensitization (80.4%), environmental tobacco smoke (51.4%), and family history of atopy (46.7%). The most frequent aeroallergen sensitization was *Dermatophagoides farinae* (59.9%), *Dermatophagoides pteronyssinus* (58.9%), and cockroaches (27.5%).

Table 1 shows the demographics and clinical characteristics of participants who attended (72; 33.9%) and did not attend (140; 66.1%) the asthma education camp. There were no significant between-group differences in baseline demographic data and baseline asthma clinical data, except that participants who attended the camp were more likely to snore (36%) than those who did not attend the camp (18%). In this study, caregiver type was categorized into three groups: mother, father, and other. Most caregivers were mothers (57.6%). The average caregiver age was 43.17 ± 10.33 years for camp attendees and 42.71 ± 10.45 years for non-attendees (*p* = 0.64). Most caregivers were aged 36–45 years (41.04%). Regarding caregiver education, 70.75% were below undergraduate level, 24.06% had a bachelor’s degree, and 5.19% had a master’s degree. The demographic data of caregivers in the two groups were similar regarding relationship to the child, caregiver age, and caregiver education.

During the 1 year follow-up, 11 and 36 children were lost to follow-up at the 3 month and 6 month visits, respectively; 165 children (77.8%) completed the 1 year visit. At the 3 month visit, the intervention group had 100% (*n* = 72) follow-up, and the control group had 92% (*n* = 129) follow-up. At the 6 month visit, the intervention group had 90% (*n* = 65) follow-up, and the control group had 79% (*n* = 111) follow-up. Both intervention and control groups had 78% (56 vs. 109) of patients with complete follow-up. Table 2 shows the results of the C-ACT and FEV1 value at each visit. At each visit, the C-ACT score did not differ significantly between the two groups. The baseline FEV1 value of attendees was 76.41 ± 13.83, and that of non-attendees was 71.28 ± 13.78 (*p* = 0.01). At the 3 month visit, the FEV1 value increased to >80%, showing that both groups had well-controlled asthma. The FEV1 value was higher in the intervention group than in the control group, although the difference was not significant (*p* = 0.38). C-ACT scores consistently increased at each follow-up visit. At the 6 month visit, the mean C-ACT score showed that the intervention group had better-controlled asthma than the control group (23.43 ± 2.96 vs. 22.79 ± 3.02, *p* = 0.22).

### 3.2. PAQLQ and Caregiver KAP Scores

PAQLQ and caregiver KAP baseline values were not significantly different between camp attendees and camp non-attendees. At the follow-up visits, there was no significant difference in QoL among pediatric asthma patients who attended camp. At the 3 month and 6 month visits, there was a significant between-group difference in caregiver KAP (16.86 ± 2.3 vs. 15.95 ± 2.78 (*p* = 0.009); 17.25 ± 2.22 vs. 16.7 ± 2.68 (*p* = 0.04)) (Table 2). The two KAP domains of treatment and exacerbation showed a significant change in mean scores from baseline (Figure 1). There was a significant overall difference in mean KAP score between the two groups (treatment *p* = 0.0036, exacerbation *p* = 0.032). At the follow-up visits, trigger item scores differed significantly between the two groups (3 month visit: 4.51 ± 0.93 vs. 4.09 ± 1.06 (*p* = 0.005); 6 month visit: 4.68 ± 0.8 vs. 4.29 ± 0.87 (*p* = 0.001); 1 year visit: 4.79 ± 0.93 vs. 4.48 ± 0.83 (*p* = 0.016)). The following factors affected KAP: child–caregiver relationship, caregiver age, and caregiver educational level (Table 3). At the 3 month visit, mothers (*n* = 118) had higher mean KAP scores than fathers (*n* = 36) or other caregivers (*n* = 49) (16.76 vs. 16.63 vs. 15.02, *p* < 0.001). Mean KAP scores were associated with educational level. Specifically, at the 3 month visit, the mean KAP score was significantly higher in the bachelor’s (18.3, *p* < 0.001) and master’s (20.46, *p* < 0.001) degree groups than in the below undergraduate level group (15.23). However, the change in KAP from the baseline was not significantly different among parent education levels at the 3 month (master’s 4.09 ± 1.44 vs. bachelor’s 3.86 ± 1.71 vs. undergraduate level 3.44 ± 1.71, *p* = 0.199), 6 month (3.72 ± 1.95 vs. 4.48 ± 2.12 vs. 4.09 ± 1.75, *p* = 0.371), and 1 year (3.5 ± 1.9 vs. 4.66 ± 1.86 vs. 4.46 ± 1.85, *p* = 0.215) follow-up visits. At all follow-up visits, older caregivers (56–69 years) had the lowest mean KAP score compared with caregivers in other age groups.

At follow-up visits, there was no statistically significant difference in level of asthma control (LOC) between children who did and did not attend an asthma education camp (Figure 2), although the proportion of controlled asthma was relatively higher in no-camp group at 3 month and 6 months. This is potentially due to the voluntary participation in the camp. Caregivers with well-controlled children may not have been interested in participating in the education camp.

The correlation between PAQLQ and level of asthma control was also investigated. Table 4 shows that the well-controlled group was significantly associated with high PAQLQ score (*p* < 0.001) at all follow-up visits.

### 3.3. KAP Scores and Level of Asthma Control

Table 5 shows the association between KAP scores and level of asthma control at each follow-up visit. At baseline, the KAP was not significantly associated with level of asthma control in both the camp and the no-camp groups. However, caregivers of well-controlled patients were more likely to have higher KAP scores than those of uncontrolled patients at each follow-up visits. Interestingly, the significant effect of KAP on the level of asthma control was observed in all follow-up visits in the no-camp group, whereas the significant effect of KAP on level of asthma control was observed at 3 month follow-up visits, although the sample size was smaller in the camp group.

## 4. Discussion

The findings show that this educational camp program improved knowledge about specific asthma-related issues, encouraged participation in physical activities, and enhanced children’s asthma management skills. Global research findings demonstrate an association between poorly controlled asthma and low QoL. Asthma educational camp programs improve asthma awareness and QoL. Children who attend asthma camps are less likely to be hospitalized, visit emergency departments, or have school absenteeism [10]. Previous research on the effects of asthma educational camps indicates that clinical status has a greater effect than QoL status [20,21,22]. Many studies evaluated the effect of asthma educational camps using measures such as pre–post C-ACT scores, pulmonary function, exercise-related dyspnea score, and analog wellbeing score [10,11]. In addition, some studies evaluated QoL on the basis of routine treatment with no camp attendance and found that good inhaler technique and good asthma control affect children’s asthma QoL [16,17]. However, this study used an intervention (an asthma educational camp) to compare the effects on QoL between the intervention and routine treatment.

There is a correlation between asthma patients’ QoL and asthma control status; patients with well-controlled asthma have better QoL [5]. Previous studies on QoL in asthmatic children in Thailand either compared QoL before and after camp attendance or compared the effect of camp attendance with routine treatment. A previous Thai study on the effects of an asthma camp found a significant improvement in asthma knowledge scores at three points: pre-camp attendance, immediately after attending the camp, and 6 months after camp attendance (mean pre- and post-test score = 24.40 vs. 27.46, *p* < 0.001). At a 6 month follow-up, QoL scores substantially improved on the emotional, activity, and symptom domains [23]. Moreover, studies assessing the effect of routine treatment show that the QoL of asthma patients is related to level of asthma control [16,17]. In our study, asthma control (assessed using the GINA guideline, the C-ACT score, and FEV1) of the children who attended the camp was relatively greater compared to those who did not attend the camp during a 1 year follow-up. Long-term follow-up can improve pulmonary function and lead to better symptom control. Thus, routine practical treatment may have a similar effect on clinical status to the educational camp. In this study, the intervention group had a higher PAQLQ score than the control group during follow-up visits. However, there were no significant differences in QoL status (PAQLQ overall *p* = 0.52, Table 2).

Caregiver knowledge is associated with asthma control. Improvement in caregivers’ KAP scores can improve management of their children’s asthma and improve medication adherence [11,24]. In this study, caregivers of the asthma-controlled group had higher knowledge scores than those of the partly controlled or uncontrolled groups. Education level affects caregivers’ ability to acquire knowledge. Higher-educated caregivers are more knowledgeable about asthma and can provide more appropriate care [24]. In this study, caregiver education had a positive relationship with KAP scores. At the first visit, caregivers who had received higher education had higher KAP scores than those with a lower educational level (Table 3). Interestingly, we observed that most caregivers answered the asthma exacerbation item on the KAP questionnaire incorrectly, suggesting that some caregivers may lack confidence in initial asthma management at home. Most caregivers expect that medical personnel are responsible for the early management of asthma exacerbation. These findings regarding caregivers’ acute asthma management were similar to those of previous related studies in Thailand [17]. We found that older caregivers were less likely to benefit from the educational program. Thus, educational programs for asthma management should be easy to access and should be tailored to and consistent with Thai culture.

One systematic review found that intervention programs for adults with asthma that encourage self-monitoring, regular medical consultation, and an asthma action plan reduce morbidity and healthcare resource use [10]. However, a meta-analysis of self-management in children showed an unclear association among self-monitoring, regular physician reviews, and an asthma action plan [10,25]. The weak correlation between clinical status and QoL may reflect measurement noise and various baseline clinical characteristics [26]. However, there is evidence that asthma self-management educational programs improve pulmonary function and self-efficacy, as well as reduce morbidity and hospitalization [10]. The present findings show that caregivers who attended camp had higher KAP scores on knowledge of disease, treatment, and management of exacerbation than those who did not attend camp (Figure 1). Clearly, educating caregivers can help to increase KAP scores, but asthma educational camps are not the only way of providing such education. Generally, physicians usually provide a short and customized education to the caregivers and patients at each follow-up visit. This continuous education given could improve caregiver’s knowledge and confidence overtime. In this study, KAP scores also increased over time among caregivers who did not attend the asthma camp, and the KAP scores were significantly associated with level of asthma control in this group (Table 5). Telemedicine is one option to deliver healthcare, especially in rural or inaccessible areas [27]. A pilot study in Italy demonstrated that a smartphone application that provides therapeutic education programs could increase asthma self-management and improve QoL in asthmatic children [28]. In a cohort trial, adolescent and adult asthma patients used a smartphone application to improve asthma control. The application was designed to reduce time-consuming inputs while offering proactive teaching and treatment support [29]. Thus, health promotion programs for parents and their children can be delivered in a variety of formats, including educational camps, small group counseling, and smartphone applications. In the COVID-19 era, smartphone applications may be used in routine practice to improve asthma self-management, asthma control, and social distancing. Children with asthma aged 6 to 18 years were home-monitored remotely for 3 months using a smartphone application and a portable spirometer. There was a significant 40% decrease in FEV1 variability, with good perception of clinical and asthma management [30].

One limitation of our study was that we could not conceal the allocation to the intervention or control groups, because all asthma patients and their caregivers were invited to attend the camp at an allergy clinic. Nonrandomization of participants can cause selection bias. It is possible that patients and their caregivers who attended the camp were more concerned about health and self-management. However, except for snoring, there were no differences in the demographic data at baseline between patients who attended the camp and those who did not attend. This quasi-experimental study could reflect the real situation, particularly in a low socioeconomic setting, where the education camp can enroll the participants only on the voluntary basis. A further study with proper randomization is suggested to confirm a causal relationship between KAP-based intervention and asthma management outcomes. Additionally, most caregivers in this study had an educational level below undergraduate level (70%), and there was no between-group difference in caregiver educational level. KAP scores were significantly different in the intervention group at the 3 month visit and the 6 month visit, although there was no between-group difference in baseline KAP scores. Although previous related studies have focused on short-term outcomes, such as KAP measurement immediately after the intervention program or 3 months after the intervention, the present study followed participants for 1 year (78% completed the 1 year follow-up) to determine the long-term effect of the intervention program. The short-term follow-up findings (e.g., at 3 months) suggest that the educational camp can improve knowledge and clinical control status. However, this effect may not persist for more than 1 year. Previous studies using asthma knowledge questionnaire surveys [15] suggested that the construction of a simple self-report asthma knowledge instrument for use as a primary outcome measure demonstrating mastery of asthma self-management skills may not be achievable. Therefore, several studies chose some KAP items, such as drug adherence or inhaler technique, to measure clinical efficacy of asthma children, such as pulmonary function or asthma control level [10,16,17]. Additionally, a further study with proper randomization is suggested to confirm a causal relationship between the KAP-based intervention and asthma management outcomes.

## 5. Conclusions

Asthma is a chronic disease that has a negative effect on patients’ QoL. Attendance at an asthma education camp can help improve self-management knowledge. The QoL of children with asthma improved if their asthma was well controlled. However, the positive effect of the education camp may last only 3–6 months after camp attendance. Therefore, to ensure effective asthma management, asthma education for children and caregivers should be a part of the routine care of asthmatic children, thus reducing the time to set up and the budget for organizing the asthma camp.

## Figures and Tables

**Figure 1 healthcare-10-01561-f001:**
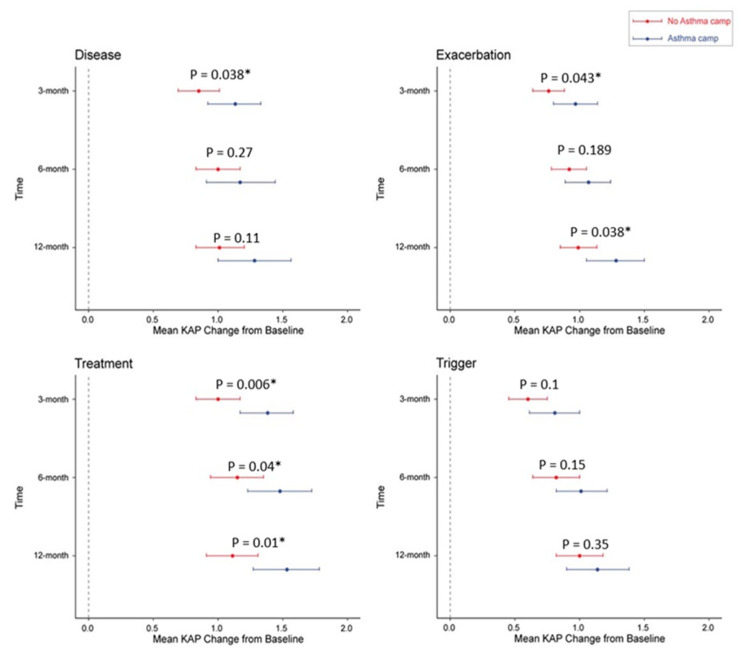
Outcome of KAP score for children who did/did not attend an asthma camp at each visit. KAP; knowledge, attitudes, and practice survey. The asterisk (*) indicates significant difference between two groups.

**Figure 2 healthcare-10-01561-f002:**
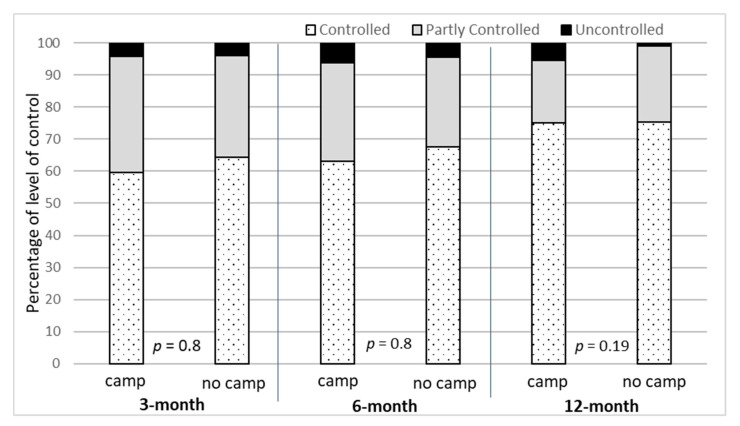
Level of asthma control for children who did/did not attend an asthma education camp.

**Table 1 healthcare-10-01561-t001:** Demographic data at baseline for patients who attended/did not attend an asthma education camp.

Variable	Attendees	Non-Attendees	*p*-Value
**Total *n* = 212**	72 (33.9)	140 (66.1)	
**Sex**			
Male	40 (55.6)	91 (65.0)	0.18
Female	32 (44.4)	49 (35.0)	
**Asthma severity**			
Mild	27 (37.5)	41 (29.3)	0.21
Moderate	38 (52.8)	74 (52.9)	
Severe	7 (9.7)	25 (17.9)	
**Comorbid diseases**			
Allergic rhinitis	36 (50.0)	63 (45.0)	0.49
Atopic dermatitis	18 (25)	26 (18.6)	0.27
Food allergy	12 (16.7)	14 (10.0)	0.16
History of acute and chronic sinusitis	8 (11.1)	6 (4.3)	0.078
History of urticarial rash	10 (13.9)	25 (17.9)	0.46
Snoring	26 (36.1)	25 (17.9)	0.003 *
Acute asthmatic attack	36 (50.0)	67 (47.9)	0.77
**Risk factors of asthma**			
Obesity	21 (29.2)	25 (17.9)	0.058
Pet owner	24 (33.3)	37 (26.4)	0.29
Environmental tobacco smoke	36 (50.0)	73 (52.1)	0.77
Family history of atopy	32 (44.4)	67 (47.9)	0.64
Aeroallergen sensitization (*n* = 204)	58 (81.7)	106 (79.7)	0.73
**Relationship to child**			
Mother	43 (59.7)	79 (56.4)	0.416
Father	15 (20.8)	23 (16.4)	
Other	14 (19.4)	38 (27.1)	
**Caregiver age**			
Age of caregiver (years) (min–max)	43.17 ± 10.33 (24–67)	42.71 ± 10.45 (26–69)	0.64
24–35	14 (19.4)	38 (27.1)	
36–45	31 (43.1)	56 (40.0)	
46–55	14 (19.4)	22 (15.7)	
56–69	13 (18.1)	24 (17.1)	
**Educational level of caregiver**			
Below undergraduate level	51 (70.8)	99 (70.7)	0.878
Bachelor’s degree	18 (25.0)	33 (25.6)	
Master’s degree	3 (4.17)	8 (5.7)	

* Significant at *p* > 0.05.

**Table 2 healthcare-10-01561-t002:** Outcomes of C-ACT score, FEV1 value, overall PAQLQ score, and overall KAP score at each visit.

Title 1	Baseline Mean ± SD	3 Months Mean ± SD	6 Months Mean ± SD	1 Year Mean ± SD	Overall *p*-Value
**C-ACT score**					
Camp	20.18 ± 3.41	22.65 ± 2.70	23.43 ± 2.96	24.05 ± 2.19	0.81
No camp	19.91 ± 3.41	22.76 ± 2.87	22.79 ± 3.02	24.01 ± 2.39	
*p*-value	0.58	0.67	0.22	0.75	
**FEV1**					
Camp	76.41 ± 13.83	88.34 ± 20.58	88.80 ± 18.99	92.33 ± 19.58	0.38
No camp	71.28 ± 13.78	84.34 ± 16.12	86.26 ± 17.06	90.76 ± 16.36	
*p*-value	0.01 *	0.165	0.29	0.24	
**PAQLQ overall score**					
Camp	5.21 ± 0.92	5.81 ± 0.81	5.91 ± 0.86	6.15 ± 0.69	0.52
No camp	5.11 ± 0.98	5.81 ± 0.85	5.93 ± 0.85	6.11 ± 0.76	
*p*-value	0.49	0.89	0.84	0.40	
**KAP overall score**					
Camp	12.65 ± 2.56	16.86 ± 2.3	17.25 ± 2.22	17.38 ± 3.00	<0.001 *
No camp	12.69 ± 2.71	15.95 ± 2.78	16.7 ± 2.68	17.00 ± 2.52	
*p*-value	0.91	0.009 *	0.04 *	0.16	

* Significant at *p* > 0.05.

**Table 3 healthcare-10-01561-t003:** Outcome of mean KAP score at baseline and 3 month visit.

	*n*	Baseline Mean KAP (95% CI)	*p*-Value *	*n*	3 Months Mean KAP (95% CI)	*p*-Value *
**Caregiver**						
Mother	122	13.26 (12.81–13.72)	Ref	118	16.76 (16.3–17.22)	Ref
Father	38	12.18 (11.37–13.00)	0.024	36	16.33 (15.5–17.16)	0.37
Other	52	11.67 (10.97–12.37)	<0.001	47	15.02 (14.3–15.75)	<0.001
**Caregiver education**						
Undergraduate	150	11.79 (11.42–12.15)	Ref	140	15.23 (14.89–15.57)	Ref
Bachelor’s	51	14.49 (13.87–15.11)	<0.001	50	18.3 (17.73–18.87)	<0.001
Master’s	11	16.36 (15.04–17.69)	<0.001	11	20.46 (19.24–21.67)	<0.001
**Caregiver age**						
24–35	52	13.17 (12.47–13.87)	Ref	49	16.49 (15.79–17.19)	Ref
36–45	87	13.05 (12.5–13.59)	0.778	86	16.8 (16.27–17.33)	0.487
46–55	36	12.47 (11.63–13.31)	0.21	33	16.36 (15.51–17.22)	0.823
56–69	37	11.32 (10.49–12.16)	<0.001	28	14.52 (13.67–15.38)	<0.001

* *p*-value from linear regression model comparing different categories of care giver’s characteristics with the reference group (Ref) at each timepoint.

**Table 4 healthcare-10-01561-t004:** PAQLQ score distribution by level of asthma control at each follow-up visit.

Level of Control	*n*	Mean PAQLQ (95% CI)	Mean Difference	*p*-Value
**3 months**				
Uncontrolled	8	4.87 (4.43–5.31)	Ref	
Partly controlled	67	5.15 (5.00–5.31)	0.29 (−0.18–0.75)	0.227
Controlled	126	6.22 (6.11–6.33)	1.36 (0.90–1.81)	<0.001 *
**6 months**				
Uncontrolled	9	4.60 (4.17–5.02)	Ref	
Partly controlled	51	5.30 (5.11–5.47)	0.70 (0.24–1.16)	0.0028 *
Controlled	116	6.31 (6.19–6.43)	1.72 (1.28–2.16)	<0.001 *
**12 months**				
Uncontrolled	4	5.01 (4.43–5.59)	Ref	
Partly controlled	37	5.41 (5.22–5.60)	0.40 (−0.21–1.01)	0.198
Controlled	124	6.38 (6.27–6.48)	1.37 (0.78–1.96)	<0.001 *

* Significant at *p* > 0.05.

**Table 5 healthcare-10-01561-t005:** Mean KAP scores by level of asthma control and asthma camp attendance at each follow-up visit.

			Mean KAP (95%CI)		*p*-Value
	*n*	Uncontrolled *	*n*	Controlled	
**Baseline**					
Camp	52	12.52 (11.82–13.22)	20	13.00 (11.87–14.13)	0.48
No camp	101	12.48 (11.95–13.00)	39	13.26 (12.42–14.10)	0.12
**3 months**					
Camp	29	16.38 (15.56–17.19)	43	17.19 (16.51–17.86)	0.13
No camp	46	15.17 (14.39–15.95)	83	16.39 (15.80–16.97)	0.015
**6 months**					
Camp	24	16.38 (15.53–17.21)	41	17.76 (17.11–18.40)	0.01
No camp	36	15.83 (14.99–16.68)	75	17.12 (16.53–17.71)	0.014
**12 months**					
Camp	14	16.86 (15.76–17.95)	42	17.90 (17.27–18.53)	0.1
No camp	27	15.78 (16.88–17.92)	82	17.40 (16.88–17.92)	0.002

* Uncontrolled group consisted of uncontrolled and partially controlled participants.

## Data Availability

The data presented in this work are available upon request from the first author.

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
