# Peer review of "Effects of an Asthma Education Camp Program on Quality of Life and Asthma Control among Thai Children with Asthma: A Quasi-Experimental Study"

_healthcare, 2022, doi:10.3390/healthcare10081561_

Round 1

Reviewer 1 Report

This is a longitudinal study on pediatric patients' treatment outcomes, using the KAP framework to differentiate an educational camp's short- and long-term effects on asthma control.  The experimental results show that the experimental group is similar to the comparison group in various personal characteristics.  The short-term or median-term effects of the intervention are documented.  Overall, this is a readable paper that has a well-organized presentation of statistical results.  However, several clarifications are needed as follows:

1. Little details on the camp were documented.  For instance, the paper should explain why and how the KAP intervention could impact on pediatric care outcomes.  What are the mechanisms that influence the outcomes of care or intervention?

2. The analysis of the KAP differential impact on outcomes should be presented. In other words, the readers are interested in learning about how K, A, and P influence the outcome of pediatric intervention in the camp.

3. The limitation section should document how future research with causal analysis could help tease out the differential effects of KAP-based intervention on healthcare outcomes.

In summary, the paper could be strengthened if the above issues are addressed by the authors. 

Author Response

The authors would like to thank the reviewers for valuable suggestions to improve our manuscript entitled “Effects of an Asthma Education Camp Program on Quality of Life and Asthma Control among Thai Children with Asthma: A Quasi-Experimental Study.”  We provided the point-by-point answer to each comment in this document.    

Reviewer #1

  1. Little details on the camp were documented. For instance, the paper should explain why and how the KAP intervention could impact on pediatric care outcomes. What are the mechanisms that influence the outcomes of care or intervention?

Answer:  We would like to thank the reviewer for suggestion. We added more details on the mechanism of asthma educational program, in terms of KAP, in the introduction part as follows:

The educational programs generally provide asthma-related knowledge such as pathology of asthma, medication use, and how to prevent an asthma attack [12]. The good attitude toward asthma management can increase confidence, drug adherence and a positive relationship with physicians; this could have an impact on the self-management practice to prevent an asthma attack [13, 14]. “  (Line 54-58)

Additionally, several studies have found no clear link between KAP and a clinical outcome.  Thus, we have added the following sentences in Section 4 to explain the KAP mechanisms that influence the clinical outcomes of children: “Previous studies of asthma knowledge questionnaires survey [15] suggested that the construction of a simple self-report asthma knowledge instrument for use as a primary outcome measure demonstrating mastery of asthma self-management skills may not be achievable. Therefore, several studies chose some KAP items, particularly, drug adherence or inhaler technique to measure clinical efficacy of asthma children such as pulmonary function or asthma control level [10,16-17].(Line 329-334)

  1. The analysis of the KAP differential impact on outcomes should be presented. In other words, the readers are interested in learning about how K, A, and P influence the outcome of pediatric intervention in the camp.

Answer:  We agreed with the reviewer that adding the impact of KAP on outcomes would adding more value to the manuscript. Therefore, we did additional analysis to show the KAP scores by level of control and camp attending status. We then added another section with Table 5 in the result part to explain these findings.

3.3. KAP scores and Level of Asthma Control

Table 5 shows the association between KAP scores and level of asthma control at each follow-up visit. At baseline, the KAP was not significantly associated with level of asthma control in both camp and non-camp group. However, caregivers of well-controlled patients were more likely to have higher KAP scores than those of uncontrolled patients at each follow-up visits. Interestingly, the significant effect of KAP on level of asthma control was observed in all follow-up visits in non-camp group. Whereas the significant effect of KAP on level of asthma control was observed at 3-month follow-up visit, although the sample size was smaller in camp group.”

Table 5. Mean KAP scores by level of asthma control and asthma camp attendance at each follow-up visit.

Mean KAP (95%CI)

p-value

N

Uncontrolled*

N

Controlled

Baseline

Camp

52

12.52 (11.82-13.22)

20

13.00 (11.87-14.13)

0.48

Non-Camp

101

12.48 (11.95-13.00)

39

13.26 (12.42-14.10)

0.12

3 months

Camp

29

16.38 (15.56-17.19)

43

17.19 (16.51-17.86)

0.13

Non-Camp

46

15.17 (14.39-15.95)

83

16.39 (15.80-16.97)

0.015

6 months

Camp

24

16.38 (15.53-17.21)

41

17.76 (17.11-18.40)

0.01

Non-Camp

36

15.83 (14.99-16.68)

75

17.12 (16.53-17.71)

0.014

12 months

Camp

14

16.86 (15.76-17.95)

42

17.90 (17.27-18.53)

0.1

Non-Camp

27

15.78 (16.88-17.92)

82

17.40 (16.88-17.92)

0.002

*Uncontrolled group consists of uncontrolled and partial controlled

  1. The limitation section should document how future research with causal analysis could help tease out the differential effects of KAP-based intervention on healthcare outcomes.

Answer:  Thank you for the suggestion. We added a sentence to suggest for future research to confirm causal relationship as follows:  “Additionally, a further study with proper randomization is suggested to confirm a causal relationship between the KAP-based intervention on asthma management outcomes.” (Line 335-336)

Reviewer 2 Report

Summary of the research and overall impression

The authors Sirasuda Sommanus, Raweerat Sitcharungsi, and Saranath Lawpoolsri present an evaluation of the effect of one-day educational camps on the quality of life of pediatric asthma patients and knowledge of the caregivers in their manuscript “Effects of an Asthma Education Camp Program on Quality of Life and Asthma Control among Thai Children with Asthma: A Quasi-Experimental Study”. The study has recruited a large cohort of pediatric asthma patients and set up a well-controlled study. The weakness of the study is the inherent bias given that caregivers could choose to attend the educational camps. Moreover, the study’s findings do not seem to align with the interpretation, as detailed below.

As the study is of general interest, it would be thematically suitable for publication in this journal and might be of interest to the readers. However, I would recommend that the authors reassess if the message of the study has been clearly highlighted.

Specific areas of improvement

-Major issues:

1) The authors conclude that asthma education should be incorporated into the routine care of pediatric asthma patients. However, the findings of the study do not seem to support that conclusion. While the knowledge of caregivers (KAP) may have been increased through the training, the quality of life of the patients did not significantly improve by this.

2) Given that the patients were not randomly allocated to the groups, there is an inherent bias to the study. The caregiver that did participate in the on-site sessions could be the more engaged caregiver and thus the one’s that would likely learn more from the training. Would all enrolled caregivers have benefited as much from the training as the “volunteer participants” if the training would have been compulsory for all enrolled parties?

3) The data does not appear to support that caregivers with a low educational background benefit most from the camps. While this can be hypothesized, the data to support this claim need to be added to the manuscript.

4) Table 4 and Figure 2 appear to indicate that the control of asthma was better in the non-camp group. Please align this finding with your description in the text.

-Minor issues:

In table 3, it is unclear, what the statistical tests are comparing. Please include a more detailed description in the legend.

Author Response

The authors would like to thank the reviewers for valuable suggestions to improve our manuscript entitled “Effects of an Asthma Education Camp Program on Quality of Life and Asthma Control among Thai Children with Asthma: A Quasi-Experimental Study.”  We provided the point-by-point answer to each comment in this document.    

Reviewer #2

-Major issues:

  1. The authors conclude that asthma education should be incorporated into the routine care of pediatric asthma patients. However, the findings of the study do not seem to support that conclusion. While the knowledge of caregivers (KAP) may have been increased through the training, the quality of life of the patients did not significantly improve by this.

Answer: We performed additional analysis (according to a suggestion from another reviewer as well) to determine the association between KAP and level of asthma control to explain this issue. Our findings suggested that the KAP scores were also increased in non-camp group (although the increase was less than the camp group). The KAP score was also associated with level of asthma control in both camp and non-camp group. We therefore added the findings in the results section and added discussion to discuss this issue as follows:

Result part:

3.3. KAP scores and Level of Asthma Control

Table 5 shows the association between KAP scores and level of asthma control at each follow-up visit. At baseline, the KAP was not significantly associated with level of asthma control in both camp and non-camp group. However, caregivers of well-controlled patients were more likely to have higher KAP scores than those of uncontrolled patients at each follow-up visits. Interestingly, the significant effect of KAP on level of asthma control was observed in all follow-up visit in non-camp group. Whereas the significant effect of KAP on level of asthma control was observed at 3-month follow-up visit, although the sample size was smaller in camp group. 

Table 5. Mean KAP scores by level of asthma control and asthma camp attendance at each follow-up visit.

Mean KAP (95%CI)

p-value

N

Uncontrolled*

N

Controlled

Baseline

Camp

52

12.52 (11.82-13.22)

20

13.00 (11.87-14.13)

0.48

Non-Camp

101

12.48 (11.95-13.00)

39

13.26 (12.42-14.10)

0.12

3 months

Camp

29

16.38 (15.56-17.19)

43

17.19 (16.51-17.86)

0.13

Non-Camp

46

15.17 (14.39-15.95)

83

16.39 (15.80-16.97)

0.015

6 months

Camp

24

16.38 (15.53-17.21)

41

17.76 (17.11-18.40)

0.01

Non-Camp

36

15.83 (14.99-16.68)

75

17.12 (16.53-17.71)

0.014

12 months

Camp

14

16.86 (15.76-17.95)

42

17.90 (17.27-18.53)

0.1

Non-Camp

27

15.78 (16.88-17.92)

82

17.40 (16.88-17.92)

0.002

*Uncontrolled group consists of uncontrolled and partial controlled

Discussion part:

“The present findings show that caregivers who attended camp had higher KAP scores on knowledge of disease, treatment, and management of exacerbation than those who did not attend camp (Figure 1). Clearly, educating caregivers can help to increase KAP scores, but asthma educational camps are not the only way of providing such education. Generally, physicians usually provide a short and customized education to the caregivers and patients at each follow-up visit. This continuous education given could im-prove caregiver’s knowledge and confidence overtime. In this study, KAP scores also increase overtime among caregiver who did not attend the asthma camp; and the KAP scores were significantly associated with level of asthma control in this group (Table 5).”   (Line 290-294)

  1. Given that the patients were not randomly allocated to the groups, there is an inherent bias to the study. The caregiver that did participate in the on-site sessions could be the more engaged caregiver and thus the one’s that would likely learn more from the training. Would all enrolled caregivers have benefited as much from the training as the “volunteer participants” if the training would have been compulsory for all enrolled parties?

Answer:  We agreed with the reviewer that non-randomization can potentially cause bias. However, we would like to conduct a study that reflect the real-world situation, particularly in a low socio-economic setting, where the education camp can enroll the participants only on the voluntary basis. We, therefore, added sentences to emphasize this strength and limitation in the discussion part as follow: “This quasi-experimental study could reflect the real situation, particularly in a low socio-economic setting, where the education camp can enroll the participants only on the voluntary basis. A further study with proper randomization is suggested to confirm a causal relationship between the KAP-based intervention on asthma management outcomes.”  (Line 315-319)

  1. The data does not appear to support the caregivers with a low educational background benefit most from the camps. While this can be hypothesized, the data to support this claim need to be added to the manuscript.

Answer:  We are sorry for our mislead explanation in the last sentence of the Abstract.  In fact, we would like to explain that although the caregivers have low education level, they can still obtain the benefit from attending the camp.  In our study, we cannot conclude that which education level get the most benefit from attending the camp as indicated in Line 189-193 that “However, the change of KAP from the baseline was not significantly different among parent education levels at ….”  

               Therefore, to avoid the misunderstanding, we deleted the phase “particularly if caregivers have low education levels” from the Abstract “Integrating asthma education into routine care could enhance asthma management in children, particularly if caregivers have low education levels

  1. Table 4 and Figure 2 appear to indicate that the control of asthma was better in the non-camp group. Please align this finding with your description in the text.

Answer:  This observation could be because of non-randomization. Caregivers with well controlled child may not be interested in participating the camp. To explain this issue, we revised the results as follows:

“At follow-up visits, there was no statistically significant difference in level of asthma control (LOC) between children who did and did not attend an asthma education camp (Figure 2). Although the proportion of controlled asthma was relatively higher in non-camp group at 3-month and 6-month. This is potentially due to the voluntary participation in the camp. Caregiver with well-controlled child may not be interested in participating in the education camp.

The correlation between PAQLQ and level of asthma control was also investigated. Table 4 shows that the well-controlled group was significantly associated with high PAQLQ score (p<0.001) at all follow-up visits.” (Line 203-209)

-Minor issues:

In table 3, it is unclear, what the statistical tests are comparing. Please include a more detailed description in the legend

Answer: Per reviewer’s suggestion, we added a footnote explaining the statistical test for this table. “* Significant at P>0.0P-value from linear regression model comparing different categories of care giver’s characteristics with the reference group at each timepoint.”

Reviewer 3 Report

The paper is drafted well,. I recommend the following corrections

1.  in abstract - make the presentation in simple english and avoid detailed technological terms and notations.

2. Introduction is good,. but needs to improve with latest findings

3. Add paper organization, in end of introduction section

4. Survey is decent,. add the following papers

Ahmed, Syed Thouheed, and Kiran Kumari Patil. "An investigative study on motifs extracted features on real time big-data signals." In 2016 International Conference on Emerging Technological Trends (ICETT), pp. 1-4. IEEE, 2016.

Syed Thouheed Ahmed, S., M. Sandhya, and S. Shankar. "ICT’s role in building and understanding indian telemedicine environment: A study." In Information and communication technology for competitive strategies, pp. 391-397. Springer, Singapore, 2019.

4. Improve the results section with a pseudo code or algorithm

5. Add an architecture diagram for ease in understanding

Author Response

The authors would like to thank the reviewers for valuable suggestions to improve our manuscript entitled “Effects of an Asthma Education Camp Program on Quality of Life and Asthma Control among Thai Children with Asthma: A Quasi-Experimental Study.”  We provided the point-by-point answer to each comment in this document.    

Reviewer #3

  1. in abstract - make the presentation in simple English and avoid detailed technological terms and notations.

Answer:  We did our best to revise the abstract part per reviewer’s suggestion. All terminology abbreviations were spell out.  

  1. Introduction is good,. but needs to improve with latest findings

Answer: We included references with the latest publications in 2020. The guideline for asthma management used and cited in this manuscript is the latest version.

  1. Add paper organization, in end of introduction section

Answer:   Our manuscript writing followed the general manuscript writing guidelines and format for medical and healthcare research. Paper organization was not usually written in the manuscript of an original research, as the manuscript has clearly specified different sections already: Introduction, Materials and Methods, Results, Discussion, Conclusion.

  1. Survey is decent,. add the following papers

Answer: Thanks for the reviewer’s suggestion. However, the first paper is irrelevant with our manuscript. [Ahmed, Syed Thouheed, and Kiran Kumari Patil. "An investigative study on motifs extracted features on real time big-data signals." In 2016 International Conference on Emerging Technological Trends (ICETT), pp. 1-4. IEEE, 2016.]

However, we have already added another paper [Syed Thouheed Ahmed, S., M. Sandhya, and S. Shankar. "ICT’s role in building and understanding indian telemedicine environment: A study." In Information and communication technology for competitive strategies, pp. 391-397. Springer, Singapore, 2019.] in our manuscript (i.e., Reference Number 27).

  1. Improve the results section with a pseudo code or algorithm

Answer: This study is healthcare research. The study design and statistical analyses were conducted according to the common acceptable research methods. We did not generate new algorithm or programming code in this study.

  1. Add an architecture diagram for ease in understanding

Answer: The study design (quasi-experimental study) used in this study was not complex. The numbers of subjects participating in the study, including lost to follow-up were specified in Table 1 and also were described in the results section. We think that the diagram might not be needed.

Round 2

Reviewer 2 Report

After addressing the review comments, the manuscript is in a shape suitable for publication.